# Mapping the colorectal cancer patient journey in Egypt: A qualitative study of diagnosis, treatment, and lifestyle perspectives

**Assem Gebreal**[1], **Karim Abdeltawab**[1], **Omar Hesham**[1], **Samr Kolkas**[1], **Somaia Khamess**[1], **Omnia Fouad**[1], **Mahmoud Ebeid**[1], **Omar Tarek**[1], **Aya Khaled**[1], **Hamza Mahmoud**[1], **Mahmoud Bassiony**[1*], **Yara Adel**[1], **Muhammed Helmi**[1], **Maryam Mansour**[1], **Omnia Nahas**[2], **Barbara Hansen**[2], **Waleed Arafat**[1], **Mona N. Fouad**[2], **Elabrar Ebrahim**[1,2], **Ahmed Ashour Badawy**[1], **Lori Brand Bateman**[2]

**1** Alexandria University, Faculty of Medicine, Alexandria, Egypt, **2** University of Alabama at Birmingham Heersink School of Medicine, Birmingham, Alabama, United States of America

* mahmoudbassiony@outlook.com

## Abstract

### Background

Colorectal cancer (CRC) in Egypt presents a significant public health challenge, ranking as the 7th most common cancer and the 8th leading cause of cancer deaths. Understanding patients' experiences is crucial to inform tailored screening and supportive care. This study aims to explore Egyptian CRC patients' experiences of diagnoses, treatment, and perceptions of a healthy diet and physical activity.

### Methods

Utilizing the Social Ecological Model (SEM) as a theoretical framework, we conducted one-hour, face-to-face, semi-structured interviews with 19 CRC patients in Alexandria, Egypt, between August and September 2023. Audio recorded interviews were transcribed and analyzed thematically to identify patterns across individual, interpersonal, organizational, social context (culture/community), and policy levels of the SEM.

### Results

The participants, 12 males and 7 females with a mean age of 54.8 (±10.2) years, predominantly came from low-income backgrounds, with the majority being either unemployed (47.4%) or disabled (31.6%). We identified facilitators and barriers to optimal care at several levels of the SEM. The main barriers were as follows: lack of awareness, symptom neglect, fear and embarrassment of colonoscopy, limited diet and activity from chemotherapy and colostomy, job loss, and poverty (individual); peers fear of colonoscopy and judgment (interpersonal); misdiagnosis, inadequate patient education,

**Data availability statement:** The data underlying this study cannot be publicly shared due to ethical restrictions, as they contain potentially identifying and/or sensitive patient information. This restriction was imposed by the Alexandria Faculty of Medicine Research Ethics Committee (the institutional body overseeing ethical compliance for this study). For researchers interested in accessing the data, requests may be directed to the committee at ethics.committee@alexmed.edu.eg.

**Funding:** This study was supported by an institutional grant from the UAB Mary Heersink Institute of Global Health at the University of Alabama at Birmingham, Birmingham, AL, USA.

**Competing interests:** The authors have declared that no competing interests exist.

**Abbreviations:** ALEX-CPC, Alexandria Society of Cancer Patient Care; AUFM, Alexandria University Faculty of Medicine; BMI, Body Mass Index; CRC, Colorectal Cancer; EMR, Eastern Mediterranean Region; EOCRC, Early-Onset Colorectal Cancer; FIT, Fecal Immunochemical Test; gFOBT, guaiac Fecal Occult Blood Test; HICs, High-Income Countries; LMICs, Low- and Middle-Income Countries; NGOs, Non-Governmental Organizations; OOP, Out-of-Pocket; SEM, Social Ecological Model; WHO, World Health Organization.

and lack of structured diet and activity programs (health organization); unsupportive work environments, cultural and religious beliefs, health literacy, and cancer stigma (social context); and unclear screening policies, lack of equipment, lack of insurance, high costs, and limited rural healthcare (policy). The main facilitators were faith resilience and positive perceptions of diet and activity (individual); family and friend support (interpersonal); positive doctor-patient relationships (health organization); NGO support (social context); and free healthcare at university hospitals (policy).

## Conclusion

The study highlights the complex interplay of barriers and facilitators CRC patients encounter throughout their experience with cancer. The findings emphasize the need for improved awareness, education, support systems; enhanced healthcare access; and targeted policy changes, especially in rural areas, to improve early diagnosis, treatment outcomes, and patient quality of life.

## 1. Introduction

Colorectal cancer (CRC) remains a major global health challenge, ranking as the third most common cancer worldwide and the second leading cause of cancer deaths, with 1.9 million new cases and 0.9 million fatalities reported in 2020 [1,2]. This burden is projected to rise to 3.2 million new cases by 2040 [1].

CRC incidence and mortality vary up to 10-fold worldwide [3]. The highest incidence is reported in high-income countries (HICs), including Australia, New Zealand, the United States (U.S.), Hungary, and Norway. The Western sedentary lifestyle, coupled with rising obesity rates and increased consumption of animal-based foods, alcohol, and tobacco, has been strongly linked to a higher risk of CRC [4,5]. However, while HICs have experienced stable or declining incidence and mortality rates for CRC in recent years, low- and middle-income countries (LMICs) are witnessing a rising trend, with rates expected to continue increasing significantly over the next few decades [3,6].

The rising rates of CRC in LMICs are presumably being driven by the adoption of Western lifestyle behaviors along with limited screening and preventive measures [6]. For instance, the Eastern Mediterranean Region (EMR) has witnessed an increasing incidence of CRC over the last two decades, ranking as the third most common cancer and the fifth leading cause of cancer death in 2022 [7,8]. However, in contrast to some European countries and the U.S., CRC survival rates in the EMR remain significantly lower, highlighting disparities in healthcare access, treatment outcomes [9,10].

In Egypt, an LMIC located in the EMR, CRC ranks as the 7th most common cancer and the 9th leading cause of cancer deaths, with 5,940 new cases and 3,096 fatalities recorded in 2022 [11]. Particularly, early-onset colorectal cancer (EOCRC), diagnosed below the age of 50, accounts for 42.4% of new cases [12,13]. In

contrast, this figure is only 10% of the new cases in HICs [14,15]. Further, the majority of CRC cases are diagnosed at a late stage, leading to an overall survival rate of two years [16].

Although remaining a significant health threat in Egypt, CRC is one of the most preventable cancers globally through screening and lifestyle modification [6]. According to a meta-analysis of eleven observational studies, screening methods, particularly colonoscopy, have been associated with a reduction in both CRC incidence and mortality by over 60% [17]. However, despite the availability of the Egyptian CRC screening guidelines, a nationwide survey revealed that only 3.2% of screening-eligible individuals were aware of the availability of CRC screening [18]. These findings indicate limited patient awareness and low adherence to established screening protocols.

Lifestyle interventions, including regular exercise and healthy eating patterns, have been shown to significantly reduce the risk of CRC [19,20]. Research has shown that high dietary fiber intake and adequate physical activity can significantly improve quality of life, lower mortality, and improve prognosis for CRC patients [21–23]. In contrast, sedentary behavior of 10 hours or more per day has been linked to a 62% increase in the risk of all-cause mortality among cancer survivors [24]. In Egypt, it is concerning that a significant majority (84.9%) of CRC patients have poor dietary patterns and are malnourished [25]. Additionally, this population reports a high prevalence of sedentary lifestyles [26]. Aly et al reported on a recent study that found 86% of CRC patients did not meet the recommendations from the World Health Organization (WHO) of at least 150 minutes of moderate-to-vigorous-intensity physical activity weekly [27]. In addition to these physical challenges, CRC patients frequently experience psychological distress, including depression, anxiety, and stress, which can further impact their quality of life and prognosis [28]. The combination of poor screening compliance and lack of optimal diet and physical activity during treatment is an alarming sign and may help explain EOCRC and the low overall survival rate among CRC patients in Egypt.

Assessing patient experiences, along with the barriers and facilitators to diagnosis, treatment, and healthy lifestyle behaviors, is essential for improving outcomes. Additionally, these insights can guide efforts to increase the acceptance and adoption of CRC screening and lifestyle changes, reduce patient distress, support coping with a cancer diagnosis and subsequent treatment, and ultimately lower incidence rates while improving survival [29,30].

To the best of our knowledge, no prior studies in Egypt have investigated CRC patients' experience with diagnosis, treatment, diet, and physical activity. Therefore, this study aims to investigate CRC patients' experiences with diagnosis, treatment, and perception of diet and physical activity in Egypt, providing insights that could inform strategies to improve care and outcomes for this vulnerable population.

## 2. Methods

### 2.1. Study design and setting

The study employed a qualitative research design through individual semi-structured interviews between August 1, 2023, and September 30, 2023. To ensure confidentiality, interviews were conducted in a private and quiet space within the Alexandria University Faculty of Medicine (AUFM) Oncology Clinic in Alexandria, Egypt. Patients were approached for interviews while waiting for their appointments at the oncology clinic.

### 2.2. Conceptual framework

This study was situated within the Social Ecological Model (SEM) framework. The SEM emphasizes multiple levels of influence and supports the idea that behaviors both affect and are affected bidirectionally between the spheres [31]. Applying the SEM can provide a comprehensive understanding of the barriers and facilitators facing CRC patients through their journey from diagnosis to survivorship. Furthermore, the SEM helped structure our findings across the various levels of the model, including the individual, interpersonal, social context (culture/community), organization, and policy levels [32].

 

## 2.3. Study participants

The population of interest was Egyptian adults with a prior diagnosis of colorectal cancer (CRC) receiving treatment at the Alexandria University Oncology Clinic. Inclusion criteria included age of at least 30 years old at the time of diagnosis and having cognitive and physical capacity to provide informed consent and actively participate in the interviews.

Medical students trained as interviewers approached patients, provided an overview of the study using the Waiting Room Script, and completed an Eligibility Screener with interested patients. Those who met the study criteria were provided with an Information Sheet to ensure their understanding of the study. Once the patient verbally consented to participate, a time, date, and location for the interview were chosen based on the participant's preference.

## 2.4. Interview duration and process

Individual semi-structured face-to-face interviews, lasting approximately one hour, were conducted with each participant. The interviews, which were audio recorded and conducted in Arabic, focused on the participants' CRC journey, including pre-diagnosis, diagnosis, treatment, and their experiences with barriers and facilitators to achieving a healthy lifestyle. The Patient Interview Topic Guide was used to guide the interview process. Additionally, a hard copy of a Demographic Questionnaire was administered by the interviewer during the interview.

## 2.5. Ethical considerations and approval

The protocol for this study was approved by the Ethics Committee of the Faculty of Medicine, Alexandria University, Egypt (I.R.B. number: 00012098, date of approval 9 July 2023). The study was conducted in accordance with the ethical standards of the Declaration of Helsinki and ethical guidelines [33]. All participants were informed that their participation was voluntary, and written informed consent was obtained from all participants prior to the interviews. Participants were provided with an Information Sheet detailing the study objectives, procedures, and their rights. Interview recordings and demographic questionnaire responses were securely stored on a password-protected computer accessible only to the lead investigator to ensure data confidentiality.

## 2.6. Data analysis

Transcripts were analyzed using thematic analysis with NVIVO 11 to facilitate the coding process [34]. A three-stage process was implemented to ensure rigor in the analysis. First, a team of 4 investigators independently read the interview transcripts to identify codes and potential themes. Second, the investigators discussed and agreed upon coding categories to develop a code book. Third, the transcripts were re-read and coded by two independent coders according to the agreed-upon coding structure. Meetings were held during the coding process to address any discrepancies and ensure consistency. Codes were elevated to themes based on recurrence and depth, where the code represents a critical insight central to our research question. Data coding continued until thematic saturation was reached, defined as the point at which no new themes or insights were identified. Finally, the investigators organized the themes into facilitators and barriers across levels of the SEM, including individual, interpersonal, social context (culture/community), organization, and policy levels.

## 3. Results

A total of 19 colorectal cancer patients participated in this study. The majority were males (n = 12, 63.2%). The mean age was 54.8 (±10.2) years, and the mean body mass index (BMI) was 21.4 (±6.4). 31.6% (n = 6) of patients were living in rural areas. Out of the 13 participants who reported their monthly income, 53.8% (n = 7) indicated that it was barely enough, 38.5%(n = 5) reported that it was not enough, and 7.7% (n = 1) considered it enough. Monthly income information was not reported by 6 participants. Most participants had a primary school education (42.1%; n = 8) or an

Industrial/Commercial/ Technical Diploma (26.3%; n = 5). Notably, most participants were unemployed (47.4%; n = 9) or disabled (31.6%; n = 6), and 100% (n = 19) were living without health insurance. Most participants (89.5%) were married and 84.2% (n = 16) were living with other family members, with a mean number of family members per household of 4.5 ± 1.7. Cancer stage data were reported for 15 participants. Among them, 66.6% (n = 10) of participants were diagnosed with stage 3 or 4.

Participant characteristics are summarized in Table 1. Our findings, guided by the SEM framework, are categorized into barriers and facilitators affecting CRC diagnosis, treatment, and lifestyle adherence for CRC patients. Each barrier or facilitator was classified to a specific level within SEM [35]. Fig 1 outlines our findings across SEM levels, with key participant quotes highlighted in S1 Table for perceived barriers and S2 Table for perceived facilitators.

### 3.1. Perceived barriers

**3.1.1. Individual level barriers.** At the individual level, lack of awareness was the most common barrier. Most participants reported neglecting symptoms such as chronic constipation and/or abdominal pain for over two years before seeking medical help, often perceiving them as minor or self-limiting. Many only sought medical help when severe symptoms emerged, such as visible rectal bleeding or complete intestinal obstruction, leading to diagnosis at an advanced cancer stage. One participant shared: *"I kept neglecting my symptoms hoping it would resolve on its own, but my daughter told me I should get myself checked"* (*male, 59 years*).

During the investigation, fear and embarrassment around colonoscopy were major barriers. Some participants discussed fear of pain, complications, not knowing the procedure, or even death from the procedure. One participant recounted, *"I kept complaining of rectal bleeding for a whole year as I was afraid to do the colonoscopy because I imagined that it would kill"* (*female, 60 years*).

Others avoided the colonoscopy due to embarrassment or discomfort. One participant shared, "*I fainted on the ground before going in. I was afraid of something entering, excuse my language, through my rectum" (female, 58 years).* This was a challenging step for some participants as it delayed their diagnosis, and they did it only when they recognized a severe worsening of their condition. One participant shared, *"I felt like I was about to die! I was experiencing severe vomiting, diarrhea, and cramps. So, I decided to do it to find out what was causing my illness"* (*female, 65 years*).

Participants also reported poor appetite, neuropathy, and back pain as side effects of chemotherapy, affecting their ability to engage in physical activities and maintain a balanced diet. One participant shared, *"After chemotherapy, I was completely having no appetite. I wanted to vomit, my stomach was hurting, and I was feeling dizzy and faint"* (*female, 41 years*).

Coping with a colostomy emerged as another significant challenge. Recurrent leak accidents caused embarrassment, affected their social interactions, negatively impacted their adherence to a healthy diet and physical activity, and ultimately diminished their quality of life. One participant recounted: "*It's a tough experience because sometimes things happen, like the waste bag bursting or leaking…It's really not a good experience at all*" (*male, 61 years*). Another added, *"I avoid eating because of the colostomy. The process of dealing with the colostomy cleaning itself, excuse my language, is not pleasant at all"* (*male, 46 years*).

The loss of employment was another significant barrier, particularly among those who worked in physically demanding jobs and could not continue working after surgery and treatment. Employers were often unsupportive. For example, a 60-year-old male stated, "*I used to work at a shop, but after this happened to me, I've been sitting at home for about three years. I haven't been able to work at all.*"

Finally, poverty created additional burdens for obtaining a timely diagnosis, accessing CRC treatment, and follow-up care. One participant explained: *"I commute from Al-Agamy, which is far, coming here to Raml Station, and that has costs. I still need dressing changes for my wound and so on, and all of this costs money. So, it has a bit of an impact as well"* (*male, 45 years*).

**Table 1. Characteristics of study participants (n = 19).**

| | | Num | % |
|---|---|---|---|
| **Age (Mean & SD)** | | **54.8 ± 10.2** | |
| **Sex** | | | |
| | Male | 12 | 63.2 |
| | Female | 7 | 36.8 |
| **BMI (Mean & SD)** | | **21.4 ± 6.4** | |
| | Underweight | 6 | 31.6 |
| | Normal weight | 10 | 52.6 |
| | Overweight | 3 | 15.8 |
| **Governorate** | | | |
| | Alexandria | 11 | 57.9 |
| | Behiera | 7 | 36.6 |
| | Gharbia | 1 | 5.3 |
| **Place of Residence** | | | |
| | Urban | 13 | 68.4 |
| | Rural | 6 | 31.6 |
| **Living Condition** | | | |
| | Living alone | 3 | 15.8 |
| | Living with family | 16 | 84.2 |
| **Number of Family Members (Mean & SD)** | | **4.5 ± 1.7** | |
| **Education level** | | | |
| | Illiterate | 4 | 21.2 |
| | Primary education | 8 | 42.1 |
| | Secondary education | 2 | 10.6 |
| | Industrial/Commercial/ Technical Diploma | 5 | 26.3 |
| **Marital Status** | | | |
| | Married | 17 | 89.5 |
| | Widowed | 2 | 10.5 |
| **Employment status** | | | |
| | Unemployed | 9 | 47.4 |
| | Disabled | 6 | 31.6 |
| | Working onsite | 3 | 15.8 |
| | Working from home | 1 | 5.3 |
| **Health Insurance Status** | | | |
| | Not medically insured | 19 | 100 |
| **Monthly Income** | (n = 13) | | |
| | Barely enough | 7 | 53.8 |
| | Not enough | 5 | 38.5 |
| | Enough | 1 | 7.7 |
| **Cancer Stage** | (n = 15) | | |
| | Stage 1 | 2 | 13.3 |
| | Stage 2 | 2 | 13.3 |
| | Stage 3 | 5 | 33.3 |
| | Stage 4 or Metastatic cancer | 5 | 33.3 |
| | Survivor | 1 | 6.7 |

BMI; Body mass index.

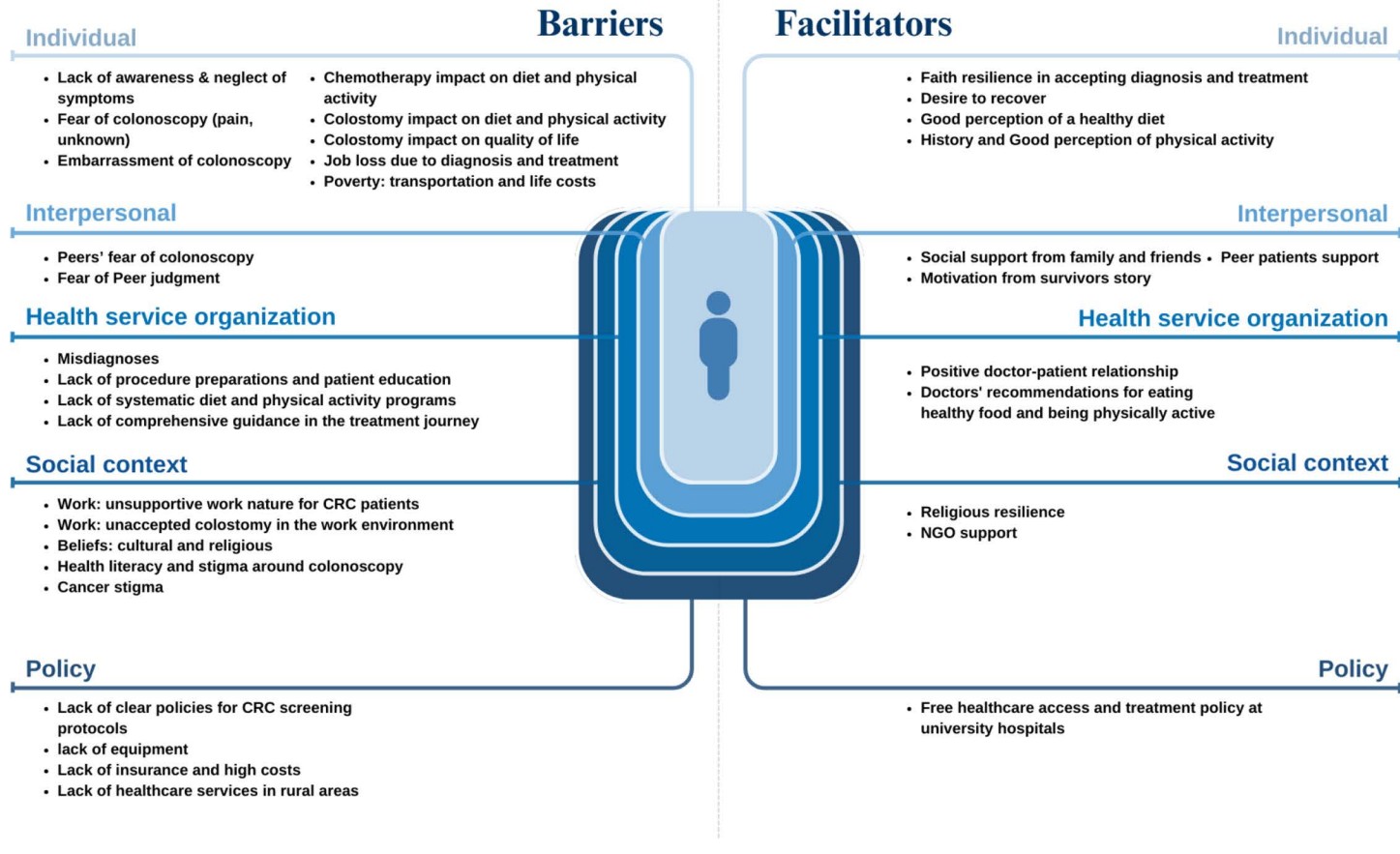

**Fig 1. Perceived barriers and facilitators at the different levels of the SEM.**

**3.1.2. Interpersonal level barriers.** At the interpersonal level, one female participant narrated that she was afraid of undergoing a colonoscopy, as some people within her social circle warned her of their bad experiences with colonoscopy. *"I know people who have had it done before (the colonoscopy), but they were scaring me about it. They kept saying that colonoscopy is very difficult"* (female, 60 years).

Additionally, another participant observed that individuals in his near circle might become apprehensive or frustrated with his health condition, particularly in the context of managing a colostomy that demands specialized care. This perception contributed to his anxiety and depression, which, in turn, limited his social interactions and heightened his fear of judgment from others. *" But also some people might be scared of me and others might be upset"* (male, 63 years).

**3.1.3. Health service organization barriers.** At this level, initial misdiagnosis was a major barrier. Many participants reported that physicians failed to suspect malignancy in early stages, often receiving diagnoses and treatment for diseases like peptic ulcers and gastritis, leading to complications such as rectal bleeding and severe weight loss. One participant recounted: *"They gave me ulcer and stomach inflammation meds, and painkillers, and none of it was helping. Eventually, I ended up not having any bowel movements for about a week and was in severe pain"* (female, 61 years).

Poor preparation for colonoscopy also discouraged patients. Some had traumatic experiences due to insufficient explanation from healthcare providers. A participant shared, *"No, no one explained anything to me about the colonoscopy...I wasn't comfortable with it. It was difficult"* (*female, 52 years*).

Moreover, while physicians advised patients to eat vegetables and a balanced diet and avoid fatty foods post-surgery, no participants received a personalized nutrition plan or referral to a nutritionist. Similarly, physical activity guidance was vague or entirely absent, and none were referred to physiotherapy for rehabilitation.

Participants also described the lack of comprehensive guidance throughout the treatment journey and expected outcomes, especially for those receiving a colostomy. This created confusion and emotional distress. A participant shared, *"I spent about twenty days feeling like a ghost in the ward. I didn't know what was wrong or what they were doing to me"* (*male, 38 years*). Another added: *"They didn't tell me anything at all...I didn't find out that I was going in for a colon tumor removal surgery until I was right in front of the operating room"* (*male, 45 years*).

**3.1.4. Social context barriers (culture/community).** At this level, several participants with stomas described the lack of workplace accommodations and stigma around visible health conditions, and insufficient support systems, which impacted their ability to find and maintain employment, as work environments were often unwelcoming or unsuitable for their condition. Concerns about leaks, odor, and frequent bag changes created practical and emotional burdens. For example, a 38-year-old male patient shared, *"I'm always worried the bag might not be sealed well, and it can leak. I could go to Siwa and work there if I didn't have this bag and catheter."*

Additionally, cultural values and religious practices intersected with stoma use. A 63-year-old male participant expressed distress over hygiene-related challenges that interfered with the ability to pray. *"I really care about hygiene…I might even stop praying because of a couple of urine spots."*

Cultural stigma, lack of community health literacy, and discomfort with colonoscopy were also mentioned frequently. Participants viewed the procedure as invasive and culturally sensitive due to its intimate nature, and they often lacked proper education about its importance. A participant explained, *"An enema and colonoscopy are very difficult, unpleasant, painful, and uncomfortable…. It's such a sensitive area"* (*male, 61 years*). Another participant added, "*What scared me was the thought of them inserting something into my rectum, and I don't know what they might do*" (*female, 60 years*).

Finally, a strong cancer-related stigma was prominent. Some participants and caregivers avoided naming cancer directly, referring to it as "the malignant disease." One participant shared, *"They told me… you got that awful, malignant disease"* (*female, 65 years*). Moreover, some caregivers hid the diagnoses of their loved ones due to fear of the psychological impact, stigma, and the belief that cancer equates to death. One said, *"At first, they didn't tell me anything. Then, eventually, they said, 'You have the malignant disease"* (*male, 60 years*).

**3.1.5. Health policy barriers.** At the policy level, participants reported barriers that hinder receiving a definitive diagnosis and treatment due to systemic health policy issues. One of the main barriers reported was the delay caused by multiple referrals between hospitals, often due to unclear policies for CRC screening protocols and a lack of imaging equipment. Another critical barrier was the lack of health insurance, which prevented many patients from affording costly diagnostic procedures in private healthcare facilities. Most of the patients were finally referred to Alexandria's main university hospital, "El-Miry Hospital," at an advanced stage with serious complications. One participant shared, *"I went to Al Agamy Hospital, where they gave me a painkiller and told me to go to Al Qabbari Hospital for an X-ray because they didn't have the equipment. At Al-Qabbari Hospital, they told me the X-ray would cost around 2000 pounds. I didn't have the money, so they referred me to El-Miry Hospital"* (*male, 45 years*).

In addition, several participants reported difficulties in accessing healthcare services, especially those living in rural areas, leading to delayed diagnosis and treatment. Long wait times and the necessity for multiple referrals compounded these delays. Another participant from a rural area mentioned: *"I came because I have a farm in Siwa and had complained several times over two years about a kidney stone. There's no healthcare there, so I went to Alexandria and had surgery here"* (*male, 39 years*).

## 3.2. Perceived facilitators

### 3.2.1. Individual facilitators.

Most participants showed resilience in accepting diagnosis and treatment due to their faith in their destiny. One participant shared, "*I am a person of faith…I accept everything that comes from God, and say, 'Alhamdulillah,' thanking Him for everything*" (*female, 61 years*). This spiritual conviction provided them with strength through their journey, fostering a belief that they were guided and protected by a higher power. Another participant added, "*For me, it's a matter of accepting that only what Allah has decreed will happen. Death is inevitable and normal. I'm prepared for it*" (*male, 38 years*).

Participants had a clear desire to recover and regain their health. Many highlighted their efforts to adhere to treatment instructions and make lifestyle changes. When asked what made it easier for them to cut out the foods they loved from their diets, one participant shared, "*I want to recover from cancer*" (*female, 60 years*).

Although most participants had no formal nutritional knowledge, the majority perceived a healthy diet as one that makes their bodies feel good after consumption. They generally considered spicy, greasy, or overly sweet foods as unhealthy, while a balanced combination of proteins, carbohydrates, and vegetables was considered healthy. This basic understanding appeared to guide many of their food choices during treatment, suggesting that this perception may facilitate adherence to a healthy diet and promote recovery. "*Healthy foods like cottage cheese, low-fat items, vegetables, fruits, fish, and lean meat… unhealthy foods are pickles, spicy foods, rich sauces, and fatty foods,*" said a 60-year-old male participant. Also, most participants mentioned that they avoid junk food and eat only homemade food. One participant suggested that, "*Anything from street vendors is not healthy. Street food is not reliable*" (*male, 45 years*).

Physical activity was defined as the ability to carry out daily tasks. Walking was the most explicitly mentioned form of exercise. One participant reflected, "*I love walking. I used to walk from Al-Asafra to Bahri—17 kilometers…It's well-known for boosting everything in the body.*" (*male, 63 years*).

Additionally, some participants believed staying active helped them better cope with treatment and recover more effectively. One participant said, "*I was used to walking a lot… the doctors also advised me to walk. It helps because I had tumors in my leg, and walking helps me with the treatment.*" (*male, 57 years*).

### 3.2.2. Interpersonal facilitators.

Social support was considered an important facilitator. Participants reported that family members, neighbors, and friends were highly supportive in terms of both financial and psychological support during the treatment journey. Participants explained that their families helped them prepare meals, ensuring their adherence to doctors' recommendations. For example, a 39-year-old male stated, "*Currently, I live with my family, where my siblings never leave me alone or in need, thank God, and my friends always visit me and keep up with how I'm doing.*" Another participant added, "*After the surgery, my eldest son's wife—may God bless her—and my daughter would prepare vegetable soup for me. Even now, they're still so supportive*" (*female, 61 years*).

Furthermore, during chemotherapy sessions and other medical procedures at the hospital, participants explained that fellow patients were cooperative and supported each other, offering encouragement and solidarity. One participant said, "*The fellow patients with me in the ward are truly well-mannered and respectful*" (*female, 60 years*).

Additionally, several participants mentioned that knowing a cancer survivor, especially a family member or close friend, highly motivated them to engage in the treatment effectively. One participant shared, "*I had a relative who had cancer and went to get treated at a hospital in Cairo. Thank God, he got married, had children, and is living a healthy life now. Everything is in God's hands and according to His will*" (*male, 60 years*).

### 3.2.3. Health service organization facilitators.

At this level, healthcare professionals, including physicians and nurses, left favorable impressions on participants in terms of dedication, support, empathy, and practicing good doctor-patient communication skills. Participants acknowledged the effort provided by medical professionals, which significantly contributed to their overall treatment experience and emotional well-being. One participant shared, "*Everyone here is respectful, and all the doctors are respectful as well*" (*male, 68 years*).

Although participants were not given individualized nutrition and physical activity plans, healthcare professionals played a vital role in encouraging patients to maintain a healthy diet and stay physically active. Most patients received guidance during treatment, emphasizing the importance of staying active and discouraging prolonged bed rest. Doctors encouraged walking, even within their homes, if they could not go outside. Additionally, they promoted the inclusion of more vegetables in their diet while advising against fatty meals and spicy foods, emphasizing their importance in maintaining proper bowel movement. One participant explained, *"The doctor who performed my surgery came to see me. He advised me to stay active but not to lift anything heavy. He also told me to eat vegetables, fruits, dairy, chicken, meat, fish, and everything"* (female, 58 years).

**3.2.4. Social context facilitators (culture/community).** The resilience demonstrated by participants is deeply influenced by the community aspects of religion. Religious teachings and practices within the community emphasize acceptance of destiny, trust in divine will, and gratitude in the face of adversity. For instance, the frequent use of phrases like "Alhamdulillah" (praise be to God) reflects a broader cultural and religious context that encourages gratitude and acceptance. This communal environment fosters and reinforces the individual beliefs that enable participants to cope with their diagnosis and treatment journey.

Additionally, several participants highlighted the role of nongovernmental organizations (NGOs) and charity organizations in supporting their treatment. The Alexandria Society of Cancer Patient Care (ALEX-CPC), a registered NGO affiliated with Alexandria University's Clinical Oncology Department, provides free cancer treatments not covered by the government and financial aid for patients who lose their jobs due to cancer. One participant explained, *"They told me to do the endoscopy with dye contrast imaging in Smouha through ALEX-CPC for free, and I did it successfully"* (female, 41 years).

**3.2.5. Health policy facilitators.** One of the main facilitators at the healthy policy level was the seamless and free access to healthcare and treatment at university hospitals, once patients were finally referred there. Patients were admitted for surgery during the same week they presented to the ER, with some even undergoing the operation on the very next day. Tumor samples were taken in the operating room, eliminating the need for a separate colonoscopy session. For example, one patient noted, *"At the El-Miry Hospital, the doctors examined me and immediately sent me to the operating room"* (male, 45 years).

Notably, none of the participants had health insurance. However, the free access to medications and chemotherapy was a smooth process with costs covered at university hospitals due to the Egyptian health policy, which was crucial for improvement, considering the social status and work status of patients. One 46-year-old male patient stated, *"I always get the chemotherapy from the hospital for free because it's not available outside, and if it is, it's very expensive."*

## 4. Discussion

This study provides a comprehensive exploration of CRC patients' journeys toward achieving a healthy lifestyle and accessing effective care in Egypt. Using the SEM, we identified barriers and facilitators faced by patients across the individual, interpersonal, organizational, social context (culture/community), and policy levels. Thematic analysis revealed a complex interconnection between barriers and facilitators during diagnosis, treatment, and post-treatment, offering insights for developing targeted interventions to support CRC patients in Egypt.

### 4.1. Perceived barriers

One key finding from the **diagnosis phase** is that more than half of the participants presented at advanced cancer stage 3 or 4 with metastases, indicating significant diagnostic delays. Studies report an estimated survival of 26 months for stage 2 CRC, and a median overall survival being 2 years [16,36]. The reasons for this phenomenon, suggested by our data and the literature, span the SEM levels.

Individual-level barriers to diagnosis identified in this study included the patient's lack of awareness and perceiving initial symptoms as minor, leading to complications, mainly rectal bleeding and intestinal obstruction. Similar barriers are

established in the literature [37–39], highlighting the urgent need to raise public awareness to counteract denial, fatalism, and coping behaviors. Furthermore, fear of colonoscopy and perceived fear of pain, lack of privacy, and embarrassment also delayed diagnosis, with patients often seeking care only after worsening symptoms or physician encouragement, consistent with previous themes in the literature [40–43].

Additionally, the stigma and embarrassment surrounding colonoscopy and digital rectal exam created a significant psychological barrier to early diagnosis. This is well-documented in studies, particularly those situated in conservative societies [44]. Poverty further restricted diagnostic access due to high costs in the private sector and a lack of equipment in accessible primary care clinics. These findings align with existing literature that individuals from LMICs face substantial cost and accessibility barriers to screening, resulting in late-stage diagnoses [45].

At the interpersonal level, the peer fear of colonoscopy among family and friends impacted patients. This aligns with findings from other studies that highlighted the lack of support from family and friends as a barrier to the uptake of colonoscopy [41].

Within the health service organization, barriers included misdiagnoses, poor preparation for procedures, and insufficient patient education. This lack of physician screening recommendations and misdiagnosis is heavily reported in the region [18,46,47], emphasizing the need for better physician communication to address patients' concerns and promote CRC screening.

At the policy level, we found a frequent lack of the application of screening protocols, particularly in rural areas. Long referral-to-diagnosis times also contributed to late-stage CRC diagnoses at tertiary care hospitals. Despite the availability of Egyptian screening guidelines, a nationwide survey reported that only 2.3% of screening-eligible participants were referred for CRC screening [18]. Similarly, Bateman et al. found that many Egyptian physicians reported screening only high-risk or symptomatic patients [47]. Therefore, training providers and expanding healthcare facilities, mobile units, and telemedicine is essential to improve timely diagnosis and treatment [48].

**During treatment**, chemotherapy side effects such as poor appetite, neuropathy, and back pain were individual-level barriers that limited participants' ability to maintain physical activity and a balanced diet. This is consistent with previous findings on chemotherapy-related functional decline [49,50]. Managing a colostomy further impaired quality of life through leaks, social isolation, and challenges with diet and physical activity. This aligns with previous studies reporting that individuals with a colostomy face significant difficulties and limitations in daily living activities, employment, and well-being [51,52]. Furthermore, poverty and employment loss, particularly in physically demanding jobs, added further treatment burdens, aligning with studies that highlight the socioeconomic challenges of cancer care in Egypt, where most patients are living in poverty [53,54].

Within the health service organization, our study identified insufficient patient education and a lack of structured programs for diet and physical activity. Although physicians advised against prolonged bed rest and emphasized eating well, detailed plans or referrals to nutritionists and physiotherapists were missing. This corroborates with the results of previous studies that highlight the lack of physician recommendations regarding what form of physical activity patients should engage in and limited access to exercise counseling as major barriers to following physical activity guidelines [49,55]. Other studies identified that malnutrition among CRC patients may worsen health and limit the effectiveness of treatments [25,56,57]. Implementing nutrition therapy during treatment is critical to enhance patient adherence to diet recommendations, prevent side effects, and shorten recovery time [58].

In the social context (culture/community), unsupportive work environments, cultural and religious beliefs, and cancer stigma present challenges for most participants. Physically demanding jobs were often unsuitable for patients, particularly for those with colostomies, further complicating their ability to maintain a stable income. Additionally, fear of colostomy leaks disrupts religious practices, such as prayer, compounding psychological distress. The lack of community-based programs to educate employers and promote inclusive hiring practices leaves many patients feeling isolated and unsupported, underscoring the need for societal changes to reduce stigma and reintegrate patients into the workforce [59].

**After treatment**, unemployment remained common. One patient reported being jobless for over three years post-diagnosis. There is a significant need for rehabilitation programs and social security support for cancer survivors [60].

## 4.2. Perceived facilitators

**At the diagnosis phase**, resilience was a key individual-level facilitator among participants. Faith and acceptance of one's destiny, influenced by religious teachings and practices within the community, were major coping mechanisms with diagnosis acceptance and treatment, consistent with previous studies that highlight the role of religious beliefs in coping with cancer [61,62]. However, mental health struggles are sometimes seen as a weakness in faith in the Arab culture and religious norms, where expressing negative emotions or discussing mental health is often taboo [63,64]. This highlights the importance of integrating Psychosocial support and culturally sensitive mental health counseling alongside spiritual resilience.

**During treatment**, participants demonstrated a positive general perception of healthy behaviors, particularly healthy diet and physical activity. Physical activity was often viewed as the ability to carry out daily tasks, with walking being the only exercise mentioned, believing it helped them better tolerate treatment and recover more effectively. This is consistent with previous studies showing that even moderate levels of physical activity can improve cancer treatment outcomes, reduce fatigue, and enhance overall quality of life [65]. At the interpersonal level, many participants highlighted the emotional and practical support they received from family, friends, and peers as a key facilitator in their journey toward treatment and recovery. Further, knowing a cancer survivor motivated patients to adhere to treatment and lifestyle recommendations. This social support has been widely recognized in the literature as a crucial determinant of health outcomes, particularly for cancer patients [66].

In the social context (culture/community), NGOs also facilitated treatment access. The healthcare system in Egypt is multifaceted, involving a wide range of public, private, and non-governmental entities that contribute to the management, financing, and delivery of medical care. NGOs and faith-based charities improve healthcare access and cancer care in both urban and rural areas through donations and services focused on raising awareness, prevention, early detection, and patient support [67,68].

Moreover, the free treatment policies at tertiary care educational hospitals were highlighted as a crucial facilitator. These provided access to urgent surgeries, chemotherapy, and oncology care funded by the government for uninsured patients. This quick transition allowed for timely consultations with oncologists and initiation of treatment compared with longer intervals experienced in developed countries [69–71]. Participants received treatment supported by government funds, indicating the accessibility and affordability of cancer care in Egypt. This is especially significant in Egypt, where the notably high poverty rate is at 29.7%, and the main healthcare financing source is out-of-pocket (OOP) expenditures, representing more than 60% of total health expenditures [72,73]. This finding highlights the importance of government support in ensuring equitable access to cancer treatment.

## 4.3. Recommendations and implications

The findings of this study highlight the urgent need for culturally tailored policy reforms, addressing barriers and facilitators to CRC diagnoses and treatment in Egypt.

To address diagnostic delays and late-stage CRC presentation, Egypt must implement culturally adapted and accessible CRC screening programs, particularly in rural areas. In 2022, Nawwar et al, developed a theory-based intervention promoting CRC screening via the low-cost, user-friendly guaiac fecal occult blood test (gFOBT) kit, distributed in a primary care unit in Egypt [74]. Following this study, the country's 100 Million Healthy Lives Campaign adopted the more sensitive fecal immunochemical test (FIT) for CRC screening, with colonoscopies covered for positive cases under the national insurance scheme [75]. However, FIT should be fully integrated into the health system, with mandatory training for healthcare providers to follow CRC screening guidelines. Healthcare providers also need medical education focused

on recognizing early CRC signs, screening guidelines, and prevention methods. A study by Khamess et al. (2023) showed that training medical students as Health Champions significantly improved their knowledge of CRC risks, screening, and treatments, suggesting this model for broader integration into Egyptian medical curricula [76]. Public health campaigns may benefit from collaborating with religious leaders, NGOs, community organizations, and media to raise awareness about CRC symptoms, promote screening, and reduce stigma around colonoscopy [77]. Engaging religious and community leaders can positively impact health behaviors among religiously conservative communities and disseminate culturally appropriate messages [78].

In treatment settings, healthcare providers should be trained to communicate effectively about treatment plans, symptom management, lifestyle changes, and Psychosocial support. It would also be beneficial for oncology care to include structured programs on diet and physical activity, with referrals to nutritionists and physiotherapists to enhance recovery and quality of life. Peer support groups for individuals undergoing treatment and post-treatment in hospitals and community centers can provide emotional support, share coping strategies, and reduce feelings of isolation. Furthermore, NGOs are critical in helping patients navigate the healthcare system, access treatment, and cope with socioeconomic challenges. Moreover, government policies that provide financial support, rehabilitation, and employment reintegration for cancer patients and survivors would be important. Survivorship programs should include ongoing mental health support, community reintegration, and partnerships with NGOs to facilitate holistic care. Additionally, further research into low-income groups and gender-specific barriers would provide valuable insights for developing targeted interventions.

### 4.4. Strengths and limitations

To the authors' knowledge, this is the first qualitative study to explore the CRC patient's journey in Egypt. Our study is distinguished by its method of performing in-person interviews with patients, which gave us valuable insights into their experiences and needs. It also addressed an important gap in knowledge about the role of diet and physical activity in Egyptian CRC patients' treatment and shed light on the barriers and facilitators to adopting a better lifestyle during and after cancer treatment.

On the other hand, this study has limitations to note. We used a convenience sample from one center, which may limit the transferability due to the broader demographic and socioeconomic diversity present within Egypt. However, the study was conducted in one of the main tertiary care hospitals located in Alexandria, the second largest city in the country, which serves a wide and diverse patient population. Additionally, semi-structured interviews rely on participants' subjective experiences and emotions. Unlike anonymous questionnaires, patients in in-person interviews may tend to express more positive aspects of their experience than negative ones; however, we tried to address that by providing questions and repeating them in different formats to obtain more comprehensive and clear answers. Throughout the study, dominance and recalcitrance did not appear to be problematic.

### 5. Conclusions

This study provides valuable insights into the lived experiences of CRC patients in Egypt. Using the SEM as a framework, our findings highlight significant barriers, including limited awareness, fear of procedures, financial constraints, and healthcare accessibility issues, while emphasizing the role of social support from family and friends, faith resilience, and free healthcare at university hospitals as facilitators. This study underscores the urgent need for culturally adapted policy reforms to improve CRC care in Egypt, including expanding fully covered CRC screening programs, awareness campaigns, and healthcare providers' training. Healthcare communication and structured support systems to address lifestyle, socioeconomic, and psychological challenges are vital to improve outcomes. These insights can help policymakers develop interventions that ultimately enhance survival rates and quality of life. Future research should explore economic, psychological, and gender-specific barriers to inform more targeted, culturally sensitive interventions.

## Supporting information

**S1 Table. Participant quotes of perceived barriers.**
(DOCX)

**S2 Table. Participant quotes of perceived facilitators.**
(DOCX)

**S1 File. Demographic questionnaire.**
(DOCX)

**S2 File. Inclusivity in global research questionnaire.**
(DOCX)

## Author contributions

**Conceptualization:** Assem Gebreal, Karim Abdeltawab, Mahmoud Bassiony, Waleed Arafat, Mona N. Fouad, Lori Brand Bateman.

**Data curation:** Assem Gebreal, Omar Hesham, Samr Kolkas, Somaia Khamess, Omnia Fouad, Omar Tarek, Aya Khaled, Hamza Mahmoud, Yara Adel, Muhammed Helmi, Maryam Mansour, Omnia Nahas.

**Formal analysis:** Assem Gebreal, Omar Hesham, Samr Kolkas, Somaia Khamess, Omnia Fouad, Mahmoud Ebeid, Omar Tarek, Aya Khaled, Hamza Mahmoud, Yara Adel, Muhammed Helmi.

**Investigation:** Assem Gebreal, Karim Abdeltawab, Omar Hesham, Samr Kolkas, Somaia Khamess, Omnia Fouad, Mahmoud Ebeid, Omar Tarek, Aya Khaled, Hamza Mahmoud, Yara Adel, Muhammed Helmi, Maryam Mansour, Omnia Nahas, Barbara Hansen, Waleed Arafat, Mona N. Fouad, Ahmed Ashour Badawy, Lori Brand Bateman.

**Methodology:** Assem Gebreal, Mahmoud Bassiony, Lori Brand Bateman.

**Project administration:** Assem Gebreal, Karim Abdeltawab, Omar Hesham, Lori Brand Bateman.

**Supervision:** Waleed Arafat, Mona N. Fouad, Lori Brand Bateman.

**Visualization:** Assem Gebreal.

**Writing – original draft:** Assem Gebreal, Karim Abdeltawab, Omar Hesham, Samr Kolkas, Omnia Fouad, Mahmoud Ebeid, Omar Tarek, Aya Khaled, Hamza Mahmoud, Mahmoud Bassiony, Yara Adel, Muhammed Helmi, Maryam Mansour, Omnia Nahas, Barbara Hansen, Lori Brand Bateman.

**Writing – review & editing:** Assem Gebreal, Karim Abdeltawab, Mahmoud Bassiony, Barbara Hansen, Elabrar Ebrahim, Lori Brand Bateman.

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
