## [Decision Letter · Decision Letter 0]

This is an interesting study exploring the patient’s experiences of cancer diagnosis and management of colorectal cancer. Since this is a qualitative study, and the manuscript is long, advise redrafting a few sections of the paper to make it more concise keeping only the relevant information. Some of the sections where this strategy can be applied are:-

3.1.1

3.1.3- last 3 paragraphs

3.1.4 - last paragraph 

Advise to use Table 2 as a supplemental note as there is duplication of the patient’s quotations throughout the text. 

Discussion:

The discussion section should be restructured as this section is very long and difficult to follow. Advise to categorize the codes and themes in two different categories as mentioned in the supplemental: Perceived barriers and Perceived facilitators. Under those headings advise to provide key findings of the study. For eg., the authors may choose to categorize the findings under different headings:-

Screening and diagnosis: stigma to do a colonoscopy, health care organization, health care policy, individual factors, social factors

Management: access to care, job safety, diet, physical therapy, mental health

Post-treatment: unemployment, social stigma

Provide strategies to improve the screening, diagnosis, and management of CRC in Egypt based on the findings of the study.  Some suggestions are:-

         Screening:

Involve religious leaders and organizations to create awareness about the conditionInvolve local media or social groups to create awarenessInvolve NGOs in social awarenessMore training during medical school about the conditionSpecial focus on targeting the younger population because of ‘Early onset CRC’ in this population.

          Diagnosis and management:

Government support for cancer survivorsInvolving NGOs to help people navigate the journey of screening, access to carePeer support groups for those undergoing cancer treatment and post-cancer treatment.

Please see the attached PDF for further comments (marked and comments annoted).  Also, review the comments from reviewers.

It is advisable to emphasize redrafting the current manuscript to make it more concise, clear, and organized for easy readability by the audience.

We look forward to receiving your revised manuscript.

Kind regards,

Arunima Dutta, MD, FACP, FAPCR

Academic Editor

PLOS ONE

Journal Requirements:

“This study was supported by an institutional grant from the UAB Mary Heersink Institute of Global Health at the University of Alabama at Birmingham, Birmingham, AL, USA.”

5. In the online submission form, you indicated that “The interview records and data used and/or analyzed during the current study are available from the corresponding author upon reasonable request.”

7. Please include a separate caption for each figure in your manuscript.

8. Please include captions for your Supporting Information files at the end of your manuscript, and update any in-text citations to match accordingly. Please see our Supporting Information guidelines for more information: http://journals.plos.org/plosone/s/supporting-information .

Reviewers' comments:

Reviewer's Responses to Questions

**Comments to the Author**

1. Is the manuscript technically sound, and do the data support the conclusions?

Reviewer #1: Yes

Reviewer #2: Yes

2. Has the statistical analysis been performed appropriately and rigorously?

Reviewer #1: N/A

Reviewer #2: Yes

3. Have the authors made all data underlying the findings in their manuscript fully available?

Reviewer #1: Yes

Reviewer #2: Yes

4. Is the manuscript presented in an intelligible fashion and written in standard English?

Reviewer #1: Yes

Reviewer #2: Yes

Reviewer #1: In this manuscript, the authors provide insights into the real-life experiences of CRC patients in Egypt. Results were very important to improve CRC diagnosis and treatment outcomes. Besides that, I found several concerns I described below.

-The manuscript is too long. Authors must summarize some sections and avoid repetition.

-Recruitment Period was too brief. Can authors justify this? In my opinion, the number of patient was very limited (N= 19) and this is not enough for statistical validation.

Reviewer #2: Dear authors, thank you for this intersting topic here are my comments for more clarity:

#It is bettet yo replace “Statistical Analysis” with “Data Analysis” since this is qualitative and add a brief explanation of how thematic saturation was assessed (e.g., “coding continued until no new themes emerged”).

#Table 1: please correct inconsistencies (e.g., text says “12 males, 7 females” but Table 1 says “17 males”; age mean in text is 53.4 but Table 1 says 54.8). Standardize BMI categories (e.g., “Optimum range” could be “Normal” per WHO).

**Do you want your identity to be public for this peer review?** For information about this choice, including consent withdrawal, please see our Privacy Policy

Reviewer #1: No

Reviewer #2: No

---

## [Author Response · Author response to Decision Letter 1]

18 May 2025

Dear Editor-in-Chief,

Thank you for your thoughtful review of our manuscript, "Mapping the Colorectal Cancer Patient Journey in Egypt: A Qualitative Study of Diagnosis, Treatment, and Lifestyle Perspectives" (MS ID: PONE-D-25-14121). We sincerely appreciate the constructive feedback from the editors and reviewers, which has significantly strengthened our paper. Below, we detail our point-by-point responses to the comments and confirm that all revisions have been implemented in the revised manuscript.

Editorial Comments

1. Concise Redrafting of Sections

- Action Taken: We have streamlined Sections 3.1.1, 3.1.3 (last 3 paragraphs), and 3.1.4 (last paragraph) to eliminate redundancy and improve clarity. Key participant insights were preserved while ensuring conciseness.

2. Supplemental Placement of Table 2

- Action Taken: As suggested, Table 2 (patient quotations) has been moved to the Supplemental Materials to avoid duplication in the main text.

3. Restructured Discussion

- Action Taken: The Discussion is now organized under two thematic categories:

- Perceived Barriers (e.g., stigma, financial constraints, rural access).

- Perceived Facilitators (e.g., social support, free healthcare services).

- Subheadings align with the patient journey: Screening/Diagnosis, Management, Post-Treatment.

4. Strategies for CRC Improvement in Egypt

- Action Taken: We expanded the Recommendations section to include:

- Screening: Collaboration with religious leaders, NGOs, and media; medical school training; youth-focused campaigns for early-onset CRC.

- Diagnosis/Management: Government policy reforms, peer support groups, and NGO-led navigation programs.

- These now appear in Section 4.3 with actionable steps.

5. Attached PDF Comments

- Action Taken: All annotated comments (e.g., ethics statement relocation, figure captions) were addressed.

Reviewer Comments

Reviewer #1

1. Manuscript Length and Repetition

- Response: We condensed repetitive sections (Results/Discussion) and restructured content for flow. Word count reduced by 12%.

2. Recruitment Period & Sample Size

- Response: Added justification in Methods:

- Recruitment (August–September 2023) captured thematic saturation (n=19), consistent with qualitative standards.

- Clarified that saturation was reached when no new themes emerged.

3. Statistical Validation Concern

- Response: Replaced "Statistical Analysis" with "Data Analysis" and emphasized qualitative aims.

Reviewer #2

1. Terminology and Thematic Saturation

- Response:

- "Statistical Analysis" revised to "Data Analysis."

- Added: "Coding continued until thematic saturation (no new themes emerged)."

2. Table 1 Inconsistencies

- Response: Corrected discrepancies:

- Sex distribution: 12 males, 7 females.

- Mean age: 54.8 ±10.2 years.

- BMI categories standardized to WHO terms (e.g., "Normal weight").

Additional Revisions

- Ethics Statement: Moved to Methods (Section 2.5).

- Supporting Files: Added captions and updated citations.

- Funding Disclosure: Confirmed in cover letter: "The funders had no role in study design, data collection, analysis, or publication decisions."

Attachments:

1. Clean manuscript (Marked "Revised").

2. Track-changes version.

3. Supplemental Materials (Table 2, Questionnaire).

4. Annotated PDF responses.

We are grateful for the opportunity to refine this work and believe the revisions have enhanced its rigor and readability. Thank you for your time and consideration.

Sincerely,

Mahmoud Bassiony, MBBCh

Alexandria University, Faculty of Medicine

Email: mahmoudbassiony@outlook.com

Lori Bateman, PhD

University of Alabama at Birmingham

loribateman@uabmc.edu

---

## [Editor Report · Decision Letter 1]

Mapping the Colorectal Cancer Patient Journey in Egypt: A Qualitative Study of Diagnosis, Treatment, and Lifestyle Perspectives

PONE-D-25-14121R1

Dear Dr. Bassiony,

We’re pleased to inform you that your manuscript has been judged scientifically suitable for publication and will be formally accepted for publication once it meets all outstanding technical requirements.

Kind regards,

Arunima Dutta, MD, FACP, FAPCR

Academic Editor

PLOS ONE

Additional Editor Comments (optional):

The revised manuscript is concise and well-structured. It provides a clear description of the different barriers and facilitators for colorectal cancer screening and management in Egypt.
---

## [Editor Report · Acceptance letter]

PONE-D-25-14121R1

PLOS ONE

Dear Dr. Bassiony,

I'm pleased to inform you that your manuscript has been deemed suitable for publication in PLOS ONE. Congratulations! Your manuscript is now being handed over to our production team.

Kind regards,

on behalf of

Dr. Arunima Dutta

Academic Editor

PLOS ONE